# Infrared Thermography—A Non-Invasive Method of Measuring Respiration Rate in Calves

**DOI:** 10.3390/ani9080535

**Published:** 2019-08-07

**Authors:** Gemma Lowe, Mhairi Sutherland, Joe Waas, Allan Schaefer, Neil Cox, Mairi Stewart

**Affiliations:** 1InterAg, Ruakura Research Centre, Hamilton 3214, New Zealand; 2School of Science, The University of Waikato, Hamilton 3216, New Zealand; 3AgResearch Ltd., Ruakura Research Centre, Hamilton 3214, New Zealand; 4Animal Inframetrics, Box 5451, Lacombe, AB, T4L 1X2, Canada; 5NeilStat Ltd., 9 Ngaere Ave, Hamilton 3210, New Zealand

**Keywords:** respiration rate, infrared thermography, flank movements, thermal fluctuations, calves, cattle health and welfare

## Abstract

**Simple Summary:**

Respiration rate (RR) is commonly used to assess states of cattle health and welfare such as pain, stress and disease. Traditionally, RR is measured by counting flank movements, a method often considered to be labour-intensive and impractical. This study investigated the use of infrared thermography (IRT) to non-invasively measure RR in calves, based on thermal fluctuations around the nostrils during inhalation and exhalation. Infrared estimates of RR were highly correlated with RR measured by observing flank movements. Through future development, the integration of IRT into existing automated systems (e.g., automated calf feeders) could support regular monitoring of calf health and welfare.

**Abstract:**

Respiration rate (RR) is a common measure of cattle health and welfare. Traditionally, measuring RR involves counting flank movements as the animal inhales and exhales with each breath. This method is often considered difficult, labour-intensive and impractical. We validated the use of infrared thermography (IRT) as an alternative method of non-invasively measuring RR in young calves. RR was simultaneously recorded in two ways: (1) by observing flank movements from video recordings; and (2) by observing thermal fluctuations around the nostrils during inhalations and exhalations from infrared recordings. For each method, the time taken to complete five consecutive breaths (a breath being a complete inhalation/exhalation cycle) was recorded and used to calculate RR (breaths/min). From a group of five calves, a total of 12 video recordings and 12 infrared recordings were collected. For each procedure, 47 sets of five consecutive breaths were assessed. The RRs measured from video recordings of flank movements and thermal fluctuations around the nostrils from infrared recordings were highly correlated (R^2^ = 0.93). Validated as a suitable method for recording RR, future research can now focus on the development of algorithms to automate the use of IRT to support its integration into existing automated systems to remotely monitor calf health and welfare.

## 1. Introduction

Respiration rate (RR) is a vital sign that can provide valuable information relating to disease [1], stress [2], pain [3] and overall cattle health and welfare [4]. Traditionally, RR is measured manually by recording the time taken for an animal to complete a specified number of breaths (e.g., the time taken to complete 10 [5] or 20 breaths [2]) or by recording the number of breaths completed in a fixed period of time (e.g., the number of breaths completed within 30 [6] or 60 s [7]); this is accomplished by observing flank movements as the animal inhales and exhales with each breath [2,7]. However, the method is often considered to be time-consuming, labour-intensive and impractical, especially in the case of long-term studies [8]. Furthermore, compared to summer conditions when rapid and heavy breathing is often observed, in cold environments flank movements are typically slower and less pronounced and, therefore, can be difficult to observe [9]. In order to accurately record flank movements, observers are also required to stand in close proximity to the animal, which may influence behavior, thereby altering RR and consequently the reliability of the results [8].

Alternative methods for recording RR have been validated, including the use of thermistors [7], infrared lasers [4], thoracic belts [10], differential pressure sensors [8] and spirometry masks [11]. However, as with flank movements, these methods have their limitations. For example, thoracic belts, spirometry masks and thermistors have to be fitted to individual morphologies and can influence natural behavior. When battery powered, these methods are also limited by battery life. Further, thoracic belts can slip from the desired position or become dislodged by other animals during the recording period, compromising the accuracy of results [8]. Similarly, thermistors (which measure thermal changes of the nostrils) can display reduced accuracy when environmental and external body surface temperatures are similar [8]. Infrared lasers offer a non-invasive means of measuring RR but, as noted by [4], they were found to be unsuitable on black cows due to their black hair absorbing the light emitted by the laser.

Infrared thermography (IRT) is another non-invasive method that has been validated for the measurement of RR in adult dairy cattle [9]. Infrared thermography detects the amount of infrared energy an object radiates; the more infrared energy, the greater the temperature of the object [12]. Infrared energy is not visible to the human eye, so infrared devices assign different colors to different levels of infrared energy to produce a false-color image visible to the human eye, known as a thermogram [13]. Stewart et al. [9] used IRT to collect recordings of the nose in order to detect thermal fluctuations associated with air movement from the nostrils during inhalation/exhalation cycles. During inhalation, cool air is drawn in from the environment, resulting in a cooling of the nostrils and a subsequent darker appearance of the nostrils in infrared recordings. In contrast, during exhalation, warm air is expelled into the environment, resulting in a warming of the nostrils and a subsequent warmer reading (and brighter appearance) of the nostrils in infrared recordings. Stewart et al. [9] recorded the time taken to complete 10 breaths (converting this into breaths per min (i.e., RR)), and found that RR measured using IRT was highly correlated with RR measured by observing flank movements in both real time and from video recordings of adult dairy cattle.

Although IRT has been validated for measuring RR in adult cattle, IRT has not been validated for measuring RR in young calves, and it should not be assumed that it would necessarily be a valid method in calves. Compared to adult cattle, both the nostrils and the amount of air displaced during inhalation and exhalation by young calves are smaller, potentially influencing the ability to accurately observe thermal fluctuations. Furthermore, as calves are more active than adult cattle, manual measures of RR become more challenging; hence, alternative methods of recording RR need to be developed. Therefore, the present study investigated the suitability of using IRT as an alternative method of non-invasively measuring RR in calves, and discusses its future potential as a technology that, through further research and development, could be integrated into automated recording systems to track the RR of calves for extended periods of time.

## 2. Materials and Methods

The current study was part of a larger project undertaken at the AgResearch Ruakura Research farm (40°44′30.822″ S, 73°59′21.508″ E) located in Hamilton, New Zealand from April to June 2017. All procedures involving animals were approved jointly by the University of Waikato Animal Ethics Committee (Protocol #1017) and the Ruakura Animal Ethics Committee (Protocol #14089). For the purposes of the present study, both video and infrared recordings were collected from a group of five Hereford calves of mixed sex (two females and three males); the animals were 27 ± 3.7 days old (range 22–32 days old) and recordings were collected on a single day within a 3.0 × 6.0 m pen (within an indoor barn).

### 2.1. Video Recordings

Video recordings were collected from calves during this period only at times whilst they were standing still, by using a hand-held video camera (HC-V270; Panasonic, Osaka, Japan). For consistency, recordings were collected by a single operator standing within the pen, 1 m in front of the calf. The field of view of the video camera was focused on the flank area in order to observe the inward and outward flank movements that occurred as the animal exhaled and inhaled, respectively. To calculate RR, video recordings were analysed by a single observer using Adobe Premiere Pro CC (version 12.0 Haberdasher; Adobe Systems, CA, USA) to play back the recordings. The observer counted the flank movements from the video and recorded the time taken for each calf to complete five breaths (a single breath being a completed inhalation/exhalation cycle).

### 2.2. Infrared Recordings

Infrared recordings were collected using a hand-held IRT camera (T650sc (accuracy: ±1.0 °C, sensitivity: <0.02 °C; resolution: 640 × 480; temperature range: −40 °C to 2000 °C; spectral range: 7.5–14 µm); FLIR systems AB, Danderyd, Sweden). Individual infrared recordings were collected at the same time as the corresponding video recordings by a second observer. For the purpose of validation, it was essential that the infrared and video recordings were collected simultaneously on the same individual to allow for a direct comparison to be made to assess the level of correlation between the two methods. Recordings were collected within an indoor barn to minimise the impact of sunlight, a factor that can influence the accuracy of infrared measurements. The IRT camera was calibrated prior to recordings being collected by entering the ambient temperature (16.5 °C), relative humidity (RH, 77%) and emissivity (ε = 0.98 (in accordance with the general known emissivity of an animal’s body [14] and as used previously for cattle [9,15])) into the camera settings. Ambient temperature and humidity were recorded using a Kestrel meter (Kestrel 3000 Pocket Weather Meter (temperature-accuracy: ±1.0 °C; resolution: ±0.1 °C; range: −29.0 °C to 70.0 °C; humidity-accuracy: ±3.0% RH; resolution: 0.1% RH; range: 5.0–95.0% RH); Nielsen-Kellerman, PA, USA).

As with the video recordings, infrared recordings were collected by a second operator standing within the pen, at a distance of 1 m in front of the calf while the calf was standing. The field of view for the infrared recordings was focused on the nose in order to observe the thermal fluctuations that occurred as air moved in and out of the nostrils as the animal inhaled and exhaled with each breath (Figure 1). For consistency and accuracy of the results, in addition to (1) minimising sunlight; (2) adjusting for environmental conditions; and (3) keeping a consistent distance from the calf, infrared recordings were collected at an angle of 90° in relation to the front of the nose. To calculate RR, infrared recordings were analysed by a single observer using Adobe Premiere Pro CC (version 12.0 Haberdasher; Adobe Systems, CA, USA) to play back the recordings. From the infrared recordings, the observer monitored the color change that occurred as a result of the thermal fluctuations around the nostrils during inhalation and exhalation in order to record the time taken to complete five breaths.

### 2.3. Calculating RR

For both video and infrared recordings, the time taken to complete five breaths was converted into the number of breaths per minute (i.e., RR) using the following equation:$$\mathrm{RR}\;\left(\mathrm{breaths}/\mathrm{minute}\right)=\left(\frac{60}{x}\right)\;\times \;y$$
where 60 is the number of seconds in a minute, *x* is the time taken to complete five breaths, and *y* is the number of breaths.

### 2.4. Additional Recording Information

From the five animals observed, a collective total of 12 video recordings and 12 IRT recordings were obtained (the number of recordings collected ranged from 1 to 4 recordings/animal with an average recording length of 58 s (range: 16–152 s per recording)). Multiple sets of five consecutive breaths were collected from each recording giving a total of 47 sets of five consecutive breaths, from which RR was calculated. Each breath was only included in a single set of breaths to prevent any individual breath being counted more than once. For the purpose of this study, series of five breaths were counted to reduce the likelihood of calves moving during recordings (as occurred frequently when 10 consecutive breaths were recorded). Using Adobe Premiere Pro CC (version 12.0 Haberdasher; Adobe Systems, CA, USA) enabled the timestamps of the recordings to be displayed. These timestamps were needed for both infrared and video recordings to ensure the same period of time was being observed across the two methods.

### 2.5. Statistical Analysis

Once the RR for each set of breaths (from both the video and infrared recordings) had been determined, analysis was carried out using Genstat (version 19; VSN International Ltd., Hemel Hempstead, UK). A regression analysis was performed in order to assess the level of agreement between the two types of recording. Bias was assessed using a Bland Altman analysis and Lin’s concordance analysis was carried out to assess the equality between the two methods. Further regression analyses were carried out to test for any slope or intercept differences across animals.

## 3. Results

Measured from video recordings, RRs were found to be highly correlated with those measured from infrared recordings (R^2^ = 0.93, *p* < 0.001, Figure 2). A Bland Altman analysis of the differences between infrared and video RR recordings plotted against the average infrared and video RR recordings showed no evidence of any change in bias across the range of values and the average bias was not significant (0.02 ± 0.255 (mean difference ± standard error of the mean (SEM))) (Figure 3). In addition, Lin’s concordance analysis showed a strong level of equality between the two methods (Qc = 0.9621, *p* < 0.001). Testing for any animal differences in the relationship, further regressions indicated that neither slope (*p* = 0.281) nor intercept (*p* = 0.999) showed any significant differences between animals, suggesting no animal dependence in the relationship between infrared and video RR recordings.

## 4. Discussion

The results suggest that IRT is a suitable method for recording the RR of young calves based on thermal fluctuations occurring as air passes through the nostrils during inhalation and exhalation. Our results were consistent with those reported by Stewart et al. [9] for adult cattle. In contrast to other techniques that have been developed to estimate RR, IRT offers a non-invasive means of recording RR. Having been successfully validated, IRT could be used to record RR in young calves to provide valuable information relating to calf health and welfare during a vulnerable stage of life. Additionally, through future research, the development of algorithms, which are necessary for the automation of IRT and integration into on-farm systems to enable RR to be recorded and analysed automatically, are within reach [9].

For calves, automated milk feeders and drinking systems provide potential platforms for integration. Such integration would enable long-term monitoring of RR and, if collected alongside other physiological and behavioral measures (e.g., feeding, drinking and lying behavior, and thermal changes), could be combined to provide highly accurate composite indicators of calf health and welfare. For example, with respect to disease onset, Lowe et al. [15] found milk consumption, duration of drinking visits, number and duration of lying bouts and thermal fluctuations (of the side and shoulder) to show changes prior to the onset of neonatal calf diarrhea (NCD), suggesting that the measures were suitable as early indicators for detecting NCD. These indicators could be combined with RR, measured using IRT, to improve the ability to monitor calf health and welfare.

Furthermore, in relation to disease, previous studies [16,17] investigating the use of IRT for early detection of bovine respiratory disease (BRD) found IRT to be capable of indicating the onset of disease based on changes in eye temperature. Schaefer et al. [16,17] found that eye temperature increased significantly in response to the onset of BRD, and these changes were found to occur several days to a week before clinical signs of disease were apparent. As with other behavioral and physiological indicators, it is possible that thermal fluctuations of the eye could be recorded alongside RR in an automated system. Recording RR alongside other behavioral and physiological measures has potential for the development of highly accurate composite measures of health and welfare.

Typically, flank movements are fairly distinct and easy to see on adult cattle, except when animals are in cold environments. Measuring RR in calves based on flank movements comes with added difficulty due to calves being more active, which makes the capture of long stationary periods difficult, compromising the accuracy of the results. Focusing instead on thermal fluctuations around the nostrils, IRT provides an alternative method for measuring RR that would help to overcome the difficulty of relying on flank movements to record RR in calves. If IRT was integrated into automated calf feeder or water systems, for example, these systems typically require calves to stand individually within a narrow chute, allowing RR to be measured whilst activity is restricted. Additionally, a number of automated calf feeders are fitted with a shutter that has to open before a calf is provided access to the teat. If integrated into an automated calf feeder, RR could be recorded before the calf is given access to the teat so that the action of suckling and the presence of warm milk does not influence the accuracy of the results.

As were considered in the current study, there are aspects of using IRT to record RR that future studies and operators need to consider to ensure accurate results are obtained. For example, for calibration purposes, changes in environmental conditions over recording periods need to be entered into the IRT camera at the time of recording or during data analysis. As part of the integration of IRT into an automated system, data pertaining to environmental variables could be obtained (e.g., using a weather station or temperature and humidity loggers) and updated automatically at specified intervals directly into the system. Infrared cameras are also sensitive to sunlight, so this needs to be considered in the environments in which IRT is to be used.

Furthermore, similar to thermistors, it is possible that in hot conditions the environmental temperatures may be similar to those of the nostrils during inhalation and exhalation, making it potentially difficult to observe thermal fluctuations in order to determine RR. This requires future investigation. However, again, if integrated into an automated system such as an automated milk feeder, these systems are generally housed indoors where exposure to sunlight and heat is minimised. The automation of IRT would also enable stationary IRT cameras to be used, as opposed to the need for observers to operate hand-held cameras, which can cause disturbances to the animal. The distance of the camera to the animal also needs to be considered and kept constant to ensure reliable results. This can be managed during the collection of both manual and automated IRT recordings by having a set distance that operators stand or IRT cameras have to be installed away from the animal. Although changes in environmental variables, distance from the animal and exposure to sunlight are all factors that could influence the results being generated whilst using IRT as a method for recording RR, with consideration, operators can act to ensure that the effect of these factors can be overcome or at least minimised.

## 5. Conclusions

In conclusion, IRT was found to be a suitable method for recording RR in calves. The validation of IRT as a method for recording RR is a necessary step to be taken before this method can be automated. Having been validated, the automation of IRT is now reliant upon the future development of algorithms to enable IRT to be used as a method to automatically record and analyse RR. Furthermore, as mentioned, there are a number of factors that can influence the results that are obtained when using IRT. Therefore, to further support the automation of this method it would be worthwhile for future research to investigate the influence of such factors (e.g., sunlight and environmental conditions) on the use of IRT as an automated method of recording RR. Automation would support the integration of IRT into existing automated systems where, alongside other measures, it could be implemented as a tool for monitoring calf health and welfare.

## Figures and Tables

**Figure 1 animals-09-00535-f001:**
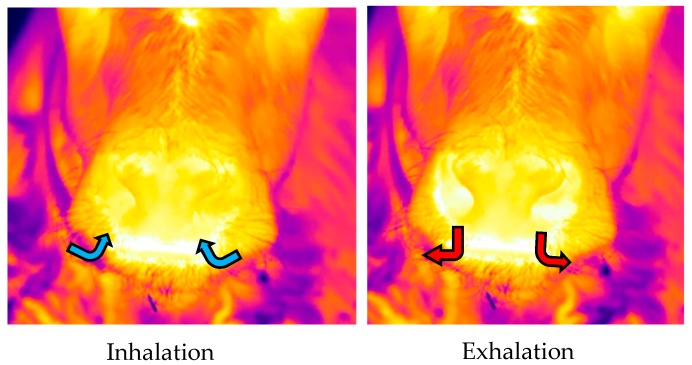
Example infrared images showing the thermal changes that occur at the nostrils during inhalation, when cold air (illustrated as blue arrows) is drawn in through the nostrils from the environment, and exhalation, when warm air (illustrated as red arrows) is expelled through the nostrils back into the environment.

**Figure 2 animals-09-00535-f002:**
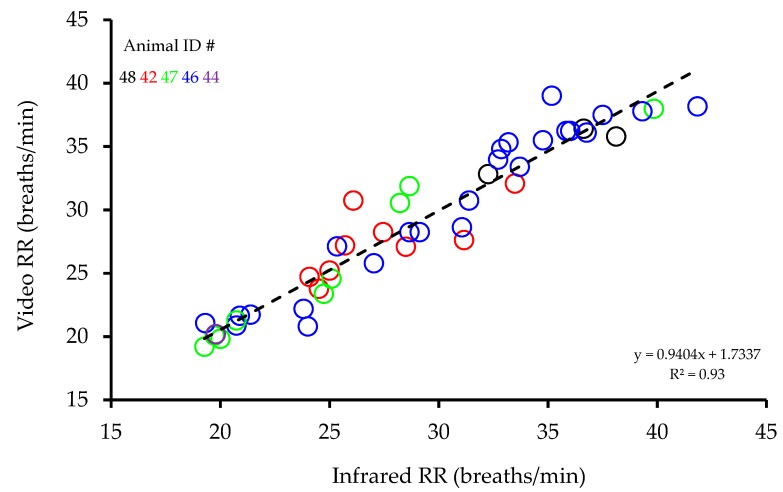
Correlation between respiration rates (RR) measured from infrared recordings and respiration rates measured from video recordings for 47 recordings.

**Figure 3 animals-09-00535-f003:**
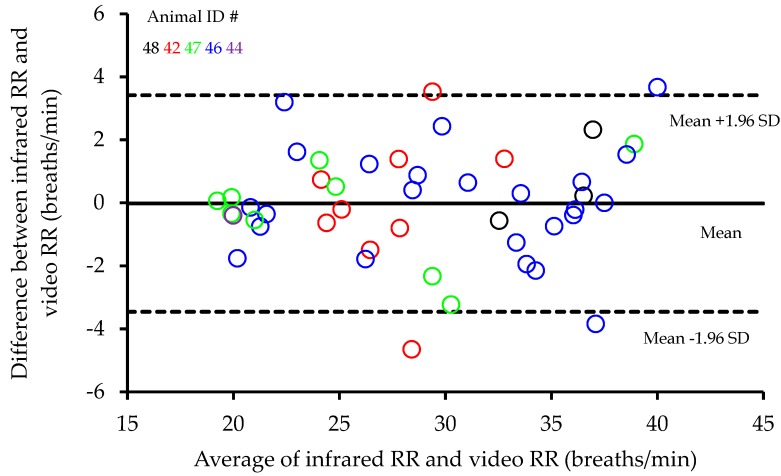
Bland Altman analysis of the average respiration rates (RR (breaths/min)) from both infrared and video recordings plotted against the differences between infrared and video recordings of RR (breaths/min).

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
