# Peer review of "Infrared Thermography—A Non-Invasive Method of Measuring Respiration Rate in Calves"

_animals, 2019, doi:10.3390/ani9080535_

Round 1
Reviewer 1 Report
I have read the Technical Note “Infrared thermography – a non-invasive method of measuring respiration rate in dairy calves”.
In general the topic of the paper is of interest in relation to health of calves. The paper is well structured and the conclusions are justified. Despite the fact that there are numerous publications in various species examining non-contact infrared thermography it is still important to assess new methodologies within these area and so far, as I know, this is the first investigation about measuring respiration rate via infrared thermography in calves. The current manuscript has novel components, is sufficiently original and the studies appear to have been carried out in a reliable and thorough scientific manner.
There are only minor points, which should be corrected. I would recommend to accept this paper after the consideration of these points.
General comments
I can’t find something about Technical Notes in the guidelines for authors. But I think the length of the paper is appropriate for this content. Therefore, I also don’t know whether it is usual to have no headings for the chapters in a Technical note for the journal “Animals”.
The emissivity selected in the present study is appropriate but the choice should be explained and justified. Can you please name the reference and write Ԑ=0.98 (line 117)? What was the accuracy (temperature range, resolution, spectral range, sensitivity) of the IRT camera (L 111) and the Kestrel meter (L 117)?
The authors should also name the location with country (USA) of a manufacturer when they name a device or software, e.g. for Adobe (line 107, 127) and for Kestrel (line 119).
Special comments for the authors
L 3 and elsewhere (L 84, 85): It is not usual to say “dairy” calves. Do you mean that they are still fed with milk (maybe suckling calves?). But the IRT can also be used in older ones. So I would only write “calves” instead of “dairy calves”
L 3: delete the full stop at the end.
L 56-57: here you can also name the RR sensor of Strutzke et al. [8]. You have only cited parts of that paper but not named the device.
L 111-114: how was it possible to get the same time span of IRT and video recordings in the analysis afterwards? Was there a time stamp in both recordings, did you use a marker or did you start both recordings via command? Or didn’t you analyze exactly the same time? Please make this clear in your text.
L 143-145: This part is difficult to understand. Please add the information how long each single video was approx.. Was it really a total of 12 video recordings (so 2-3 recordings per animal) or 12 per animal? Why not always 3 recordings per animal? I think it would become clearer if you write the number of recordings per animal and the length of the videos.
L 152: I would start this sentence in the next line to show that this is now the section “results”.
L 188 and 190: write [15,16] instead of [15-16].
L 192-195: sentence sounds incorrect (Maybe “As it is possible with other …”).
Only as a hint for the authors (L 225-230): Maybe the study of Jiao et al. (2016. Compensation method for the influence of angle of view on animal temperature measurement using thermal imaging camera combined with depth image, http://dx.doi.org/10.1016/j.jtherbio.2016.07.021) can be helpful to minimize problems. But I have no experiences with this method.
L 249-250: layout is different from the following lines
L 253: Isn’t it Schütz (instead of Schutz)?
Author Response
Thank you for your valuable suggestions on this manuscript. Please see our responses outlined below.
Reviewer 1
Comments and Suggestions for Authors
I have read the Technical Note “Infrared thermography – a non-invasive method of measuring respiration rate in dairy calves”.
In general the topic of the paper is of interest in relation to health of calves. The paper is well structured and the conclusions are justified. Despite the fact that there are numerous publications in various species examining non-contact infrared thermography it is still important to assess new methodologies within these area and so far, as I know, this is the first investigation about measuring respiration rate via infrared thermography in calves. The current manuscript has novel components, is sufficiently original and the studies appear to have been carried out in a reliable and thorough scientific manner.
There are only minor points, which should be corrected. I would recommend to accept this paper after the consideration of these points.
General comments
I can’t find something about Technical Notes in the guidelines for authors. But I think the length of the paper is appropriate for this content. Therefore, I also don’t know whether it is usual to have no headings for the chapters in a Technical note for the journal “Animals”.
AU: We have made mention of this to the editor, if necessary we will reformat the style of the manuscript with the addition of key headings.
The emissivity selected in the present study is appropriate but the choice should be explained and justified. Can you please name the reference and write Ԑ=0.98 (line 117)? What was the accuracy (temperature range, resolution, spectral range, sensitivity) of the IRT camera (L 111) and the Kestrel meter (L 117)?
AU: Specifications for both the IRT camera (L116-118) & the Kestrel meter (L127-129) have now been included. ‘Ԑ=’ has been added ahead of the emissivity value (L 125) and references have been provided to acknowledge this value falls within the known emissivity of animals and is similar to the emissivity values of other studies on cattle (L125-126).
The authors should also name the location with country (USA) of a manufacturer when they name a device or software, e.g. for Adobe (line 107, 127) and for Kestrel (line 119).
AU: USA added to equipment/software details as suggested (L 112,129 & 146).
Special comments for the authors
L 3 and elsewhere (L 84, 85): It is not usual to say “dairy” calves. Do you mean that they are still fed with milk (maybe suckling calves?). But the IRT can also be used in older ones. So I would only write “calves” instead of “dairy calves”
AU: Changed as suggested throughout the manuscript.
L 3: delete the full stop at the end.
AU: Full stop has been removed from the title (L3).
L 56-57: here you can also name the RR sensor of Strutzke et al. [8]. You have only cited parts of that paper but not named the device.
AU: This reference has been included amongst other references of examples of alternative methods which have been developed for recording RR (L62).
L 111-114: how was it possible to get the same time span of IRT and video recordings in the analysis afterwards? Was there a time stamp in both recordings, did you use a marker or did you start both recordings via command? Or didn’t you analyze exactly the same time? Please make this clear in your text.
AU: Timestamps were used on the different recordings to ensure the same time periods were being observed. This has been clarified within the manuscript. We have now added that “Timestamps were used for both infrared and video recordings to ensure the same period of time was observed across the two different methods” (L169-170).
L 143-145: This part is difficult to understand. Please add the information how long each single video was approx.. Was it really a total of 12 video recordings (so 2-3 recordings per animal) or 12 per animal? Why not always 3 recordings per animal? I think it would become clearer if you write the number of recordings per animal and the length of the videos.
AU: To clarify it has been stated “From the 5 animals observed, a collective total of 12 video recordings and 12 IRT recordings were obtained (the number of recordings collected ranged from 1-4 recordings/animal with an average recording length of 58 seconds (range: 16-152 seconds) per recording” (L162-164)
L 152: I would start this sentence in the next line to show that this is now the section “results”.
AU: This sentence has been shifted down a line to begin the start of the next paragraph to differentiate between the statistical analysis and results sections (L178)
L 188 and 190: write [15,16] instead of [15-16].
AU: Changed as suggested. Note these numbers have changed due to the addition of references earlier in the manuscript since initial submission.
L 192-195: sentence sounds incorrect (Maybe “As it is possible with other …”).
AU: A comma has been added to help clarify this sentence (L 229)
Only as a hint for the authors (L 225-230): Maybe the study of Jiao et al. (2016. Compensation method for the influence of angle of view on animal temperature measurement using thermal imaging camera combined with depth image, http://dx.doi.org/10.1016/j.jtherbio.2016.07.021) can be helpful to minimize problems. But I have no experiences with this method.
L 249-250: layout is different from the following lines
AU: We believe this refers to the slight difference in line spacing following the first reference listed. This has been adjusted for consistency with the rest of the reference list (L 289).
L 253: Isn’t it Schütz (instead of Schutz)?
AU: Corrected (L293)
Reviewer 2 Report
I am not convinced there is sufficient account taken of animal differences. There were only 5 animals used and over half the observations were form just one animal (46). Was there something specific about this animal that made measurements easier? For example coat, degree of flank movement, variation in the FLIR reading between inhaled and exhaled breath. Would the correlation have been the same if the same number of observations were taken from each animal? I have to assume that the difficulty in obtaining a measure with both methods was higher for some animals than others. This maybe an inherent flaw of the method that the current trial was unable to elucidate. This invalidates the results.
You state on line 28 and in the title that the method was for dairy calves but on line 99 you indicate that Hereford calves of mixed sex were actually used.
Author Response
Thank you for your comments on this manuscript. Please see responses outlined below.
I am not convinced there is sufficient account taken of animal differences. There were only 5 animals used and over half the observations were form just one animal (46). Was there something specific about this animal that made measurements easier? For example coat, degree of flank movement, variation in the FLIR reading between inhaled and exhaled breath. Would the correlation have been the same if the same number of observations were taken from each animal? I have to assume that the difficulty in obtaining a measure with both methods was higher for some animals than others. This maybe an inherent flaw of the method that the current trial was unable to elucidate. This invalidates the results.
AU: The multiple measurements collected per individual animal are presented in the Bland Altman analysis in Figure 3. As discussed in L184-187 “to test for any animal differences in the relationship, further regression analyses were carried out to test for any slope or intercept differences for all animals. Neither slope (P=0.281) nor intercept (P=0.999) showed any significant differences between animals, suggesting no animal dependence in the relationship between infrared and video RR recordings”. With no animal dependence it is likely the correlation would be the same had the same number of observations been collected per animal.
There was nothing particular about animal #46 that made it more suitable for recordings, other than that due to the length of recordings being longer for this animal it would have been standing still for a greater amount of time than other animals and therefore more breaths were observed from the recordings. The range of the number of recordings collected per animal and the average recording length has been added in L162-164.
You state on line 28 and in the title that the method was for dairy calves but on line 99 you indicate that Hereford calves of mixed sex were actually used.
AU: No longer referred to as “dairy calves” throughout manuscript.
Reviewer 3 Report
The use of remote sensing technology for animal monitoring is an increasing area of interest. Technologies need to be validated before they can be developed for practical application. Therefore the subject of this Technical Note is of interest.
The use of infrared thermography to measure respiration rate has been validated in several species, including dairy cows and is well accepted. The challenge of this technique is mostly in the practical application. When conducting a validation study it is important to be very clear about the methodology and particularly about the data processing. I have some concern that it is not clear how the analysed data was selected. The number of breaths counted were only 5, rather than the more accepted 10 breath. Recordings were collected within a 2 h period, but only 8 minutes were analysed. It is important to report on clear criteria why data was excluded to avoid cherry picking, which may result in seemingly good correlation that is not repeatable.
Multiple data was collected from the same animals, so correct statistical analysis should be used to take this into account (for example use Bland Altman analysis with multiple measurements per subject).
Before IRT is suitable for automated measures of RR additional validation of exclusion factors (such as light and movement) would be needed. The limitations of this study should be recognised in the conclusions.
Author Response
Thank you for your comments from reviewing this manuscript. Please see responses as outlined below.
The use of remote sensing technology for animal monitoring is an increasing area of interest. Technologies need to be validated before they can be developed for practical application. Therefore the subject of this Technical Note is of interest.
The use of infrared thermography to measure respiration rate has been validated in several species, including dairy cows and is well accepted. The challenge of this technique is mostly in the practical application. When conducting a validation study it is important to be very clear about the methodology and particularly about the data processing. I have some concern that it is not clear how the analysed data was selected. The number of breaths counted were only 5, rather than the more accepted 10 breath. Recordings were collected within a 2 h period, but only 8 minutes were analysed. It is important to report on clear criteria why data was excluded to avoid cherry picking, which may result in seemingly good correlation that is not repeatable.
Au: To clarify, recordings were only collected within the 2 hour observation period when calves were standing still, as opposed to being collected for a total recording duration of 2 hours. We have removed the statement that recordings were collected within a 2 hour period and have instead included the number of recordings collected per animal and average recording length (L 162-164). Unfortunately, as acknowledged in the manuscript, although we had initially set out to record the time taken to complete 10 breaths this was not possible due to activity of the calves; we therefore had to make the decision to record the time taken to complete 5 breaths instead.
Multiple data was collected from the same animals, so correct statistical analysis should be used to take this into account (for example use Bland Altman analysis with multiple measurements per subject).
AU: The multiple measurements collected per individual animal are presented in the Bland Altman analysis in Figure 3. As discussed in L184-187 “to test for any animal differences in the relationship, further regression analyses were carried out to test for any slope or intercept differences for all animals. Neither slope (P=0.281) nor intercept (P=0.999) showed any significant differences between animals, suggesting no animal dependence in the relationship between infrared and video RR recordings”.
Before IRT is suitable for automated measures of RR additional validation of exclusion factors (such as light and movement) would be needed. The limitations of this study should be recognised in the conclusions.
AU: It has been noted in the conclusion that to further support the automation of IRT for the purpose of recording RR, future studies could investigate factors which may influence the suitability of this method (L269-278).